# Meta-analysis reveals an extreme "decline effect" in the impacts of ocean acidification on fish behavior

**Jeff C. Clements**[1¤]*, **Josefin Sundin**[1,2], **Timothy D. Clark**[3], **Fredrik Jutfelt**[1]*

1 Department of Biology, Norwegian University of Science and Technology, Trondheim, Norway,
2 Department of Aquatic Resources, Swedish University of Agricultural Sciences, Drottningholm, Sweden,
3 School of Life and Environmental Sciences, Deakin University, Geelong, Australia

¤ Current address: Fisheries and Oceans Canada, Moncton, Canada
* jeffery.clements@dfo-mpo.gc.ca, jefferycclements@gmail.com (JCC); fredrik.jutfelt@ntnu.no (FJ)

**Data Availability Statement:** All statistical results, annotated R code and raw data (including full dataset, original data files uploaded to R for analyses, and source data for figures) are available in the Supporting information.

## Abstract

Ocean acidification—decreasing oceanic pH resulting from the uptake of excess atmospheric $CO_2$—has the potential to affect marine life in the future. Among the possible consequences, a series of studies on coral reef fish suggested that the direct effects of acidification on fish behavior may be extreme and have broad ecological ramifications. Recent studies documenting a lack of effect of experimental ocean acidification on fish behavior, however, call this prediction into question. Indeed, the phenomenon of decreasing effect sizes over time is not uncommon and is typically referred to as the "decline effect." Here, we explore the consistency and robustness of scientific evidence over the past decade regarding direct effects of ocean acidification on fish behavior. Using a systematic review and meta-analysis of 91 studies empirically testing effects of ocean acidification on fish behavior, we provide quantitative evidence that the research to date on this topic is characterized by a decline effect, where large effects in initial studies have all but disappeared in subsequent studies over a decade. The decline effect in this field cannot be explained by 3 likely biological explanations, including increasing proportions of studies examining (1) cold-water species; (2) nonolfactory-associated behaviors; and (3) nonlarval life stages. Furthermore, the vast majority of studies with large effect sizes in this field tend to be characterized by low sample sizes, yet are published in high-impact journals and have a disproportionate influence on the field in terms of citations. We contend that ocean acidification has a negligible direct impact on fish behavior, and we advocate for improved approaches to minimize the potential for a decline effect in future avenues of research.

## Introduction

Publications presenting new hypotheses or groundbreaking scientific discoveries are often followed by attempts to replicate and build upon the initial research. In many instances, however, follow-up studies fail to replicate initial effects and/or report smaller effect sizes. The tendency

**Funding:** This work was supported by a Marie Skłodowska-Curie Individual Fellowship funded through the European Union Horizon 2020 program (project number 752813 to J.C.C.), the Australian Research Council's Future Fellowship program (FT180100154 to T.D.C.), and the Research Council of Norway (262942 to F.J.). The funders had no role in study design, data collection and analysis, decision to publish, or preparation of the manuscript.

**Competing interests:** Three of the authors (J. Sundin, T. Clark, and F. Jutfelt) have previously raised concerns about, and have requested formal investigations into, the scientific integrity of some studies published by Drs. Philip Munday and Danielle Dixson.

for initial scientific findings—which can show strong effects with large effect sizes—to lose strength over time is referred to as the "decline effect" [1]. This phenomenon was first described in the 1930s and has since been documented in a range of scientific disciplines [1], including ecology and evolution [2,3]. It captures the concept of initial reports with large effect sizes that overestimate reality. In such instances, the early, large effect sizes are the key problem, not the subsequent decline. The decline effect could therefore equally be referred to as the "early inflation effect." Nonetheless, decline effects can be problematic by delaying accurate scientific understanding of a given phenomenon and can have applied ramifications, for example, to policy making [4].

Over the past 15 years, biologists have documented substantial impacts of ocean acidification on marine biota [5]. With more than 300 papers published per year from 2006 to 2015, the exponential growth of ocean acidification studies represents one of the fastest expanding topics in the marine sciences [6] and underscores the perceived risk of ocean acidification to ecosystem resilience. In recent years, however, there has been increasing skepticism and uncertainty around the severity of ocean acidification effects on marine organisms [6,7].

Some of the most striking effects of ocean acidification are those concerning fish behavior, whereby a series of sentinel papers in 2009 and 2010 published in prestigious journals reported large effects of laboratory-simulated ocean acidification [8–10]. Since their publication, these papers have remained among the most highly cited regarding acidification effects on fish behavior. The severe negative impacts and drastic ecological consequences outlined in those studies were highly publicized in some of the world's most prominent media outlets [11–13] and were used to influence policy through a presentation at the White House [14]. Not only were the findings alarming, but the extraordinarily clear and strong results also left little doubt that the effects were real, and a multimillion dollar international investment of research funding was initiated to quantify the broader impacts of ocean acidification on a range of behaviors. In recent years, however, an increasing number of papers have reported a lack of ocean acidification effects on fish behavior, calling into question the reliability of initial reports.

Here, we present a striking example of the decline effect over the past decade in research on the impact of ocean acidification on fish behavior. We find that initial effects of acidification on fish behavior have all but disappeared over the past 5 years and present evidence that common biases influence reported effect sizes in this field. Ways to mitigate these biases and reduce the time it takes to reach a "true" effect size, broadly applicable to any scientific field, are discussed.

## Results and discussion

### Declining effects

Based on a systematic literature review and meta-analysis ($n$ = 91 studies), we found evidence for a decline effect in ocean acidification studies on fish behavior (Fig 1a and 1b). Generally, effect size magnitudes (absolute lnRR) in this field have decreased by an order of magnitude over the past decade, from mean effect size magnitudes >5 in 2009 to 2010 to effect size magnitudes <0.5 after 2015 (Fig 1a and 1b, S1 Table). Mean effect size magnitude was disproportionately large in early studies, hovered at moderate effect sizes from 2012 to 2014, and has all but disappeared in recent years (Fig 1a and 1b).

The large effect size magnitudes from early studies on acidification and fish behavior are not present in the majority of studies in the last 5 years (Fig 1b, S1 Table). This decline effect could be explained by a number of factors, including biological. For example, cold-water fish in temperate regions experience a higher degree of temporal variability in carbonate chemistry parameters over large spatial areas [15]. Therefore, they may be less sensitive to changes in

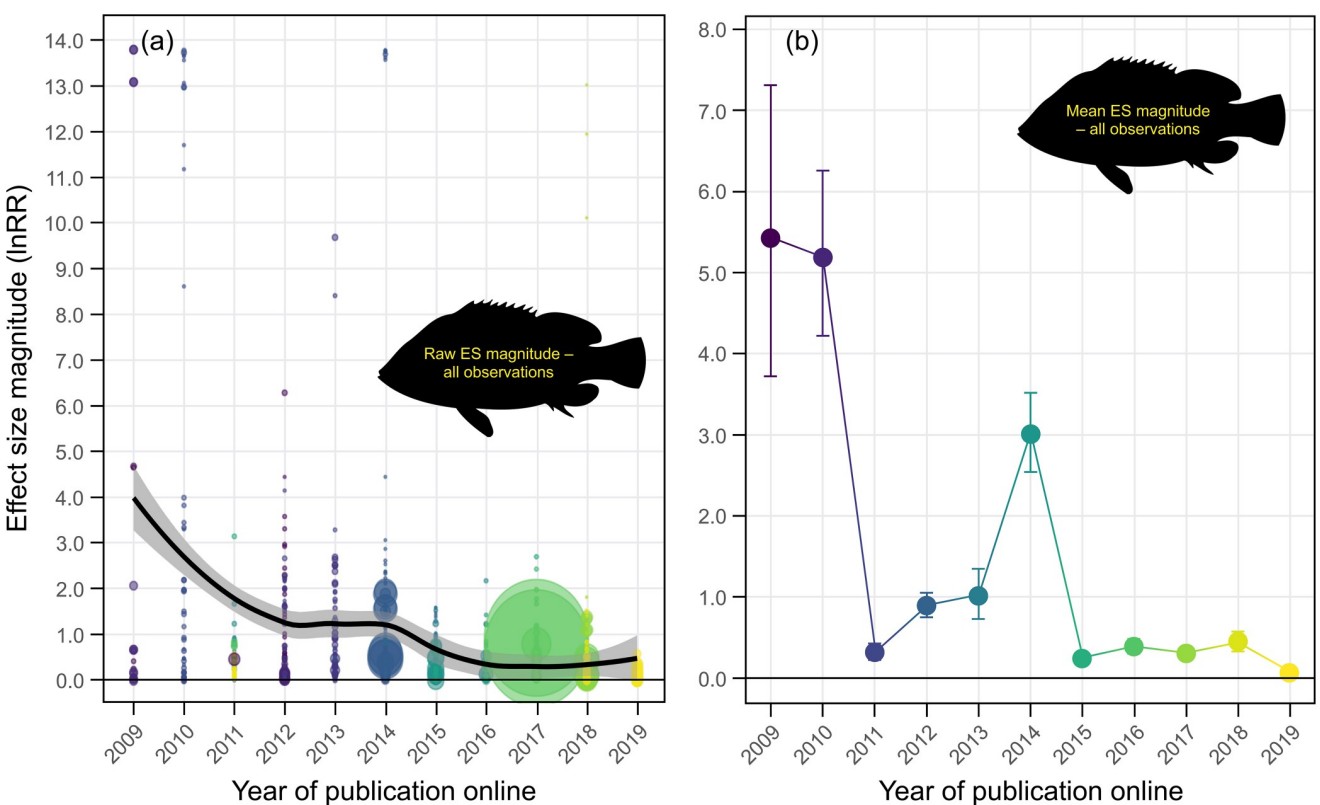

**Fig 1. The decline effect in ocean acidification research on fish behavior. (a)** Trend in raw effect size magnitudes (absolute lnRR) for each experiment in our dataset plotted as a function of year of publication online and color coded according to study. Data are fit with a Loess curve with 95% confidence bounds. **(b)** Mean effect size magnitude (absolute lnRR ± upper and lower confidence bounds) for each year of publication (online) in our dataset. Mean effect size magnitudes and confidence bounds were estimated using Bayesian simulations and a folded normal distribution. Note: Colors for **(b)** are aesthetic in nature and follow a gradient according to year of publication. Source data for each figure panel can be found in S1 Data. ES, effect size.

seawater $CO_2$ as per the Ocean Variability Hypothesis [16]. As such, if an increasing number of studies on cold-water species over time was responsible for the decline effect, removing cold-water species from the dataset (i.e., only including warm-water species) should result in the decline effect trend disappearing. This was not the case, as the decline effect persisted when only warm-water species were considered (Fig 2a). In the same vein, the strongest ocean acidification effects on fish behavior have undoubtedly been reported for chemical cue (herein "olfactory") responses, and an increasing number of studies on nonolfactory behaviors could explain the decline effect. If this was true, removing nonolfactory behaviors from the dataset should negate the decline effect trend. Again, this was not the case (Fig 2b). Finally, early studies of ocean acidification and fish behavior used larval fish, which are typically considered to be more sensitive to environmental perturbations than juveniles and adults. If a greater proportion of studies used less sensitive life stages through time, then removing those life stages and focusing exclusively on larvae should abolish the decline effect. Once again, this was not the case (Fig 2c). These analyses show that ocean acidification studies on fish behavior exhibit a decline effect that is not explainable by 3 biological processes commonly considered important drivers of acidification effects (Fig 2a–2c, S1 Table).

While we were able to test and exclude 3 biological factors, there are other potential factors that could drive the decline which are not readily testable from our database. For example, while we were able to partially test for the influence of background $CO_2$ variability by

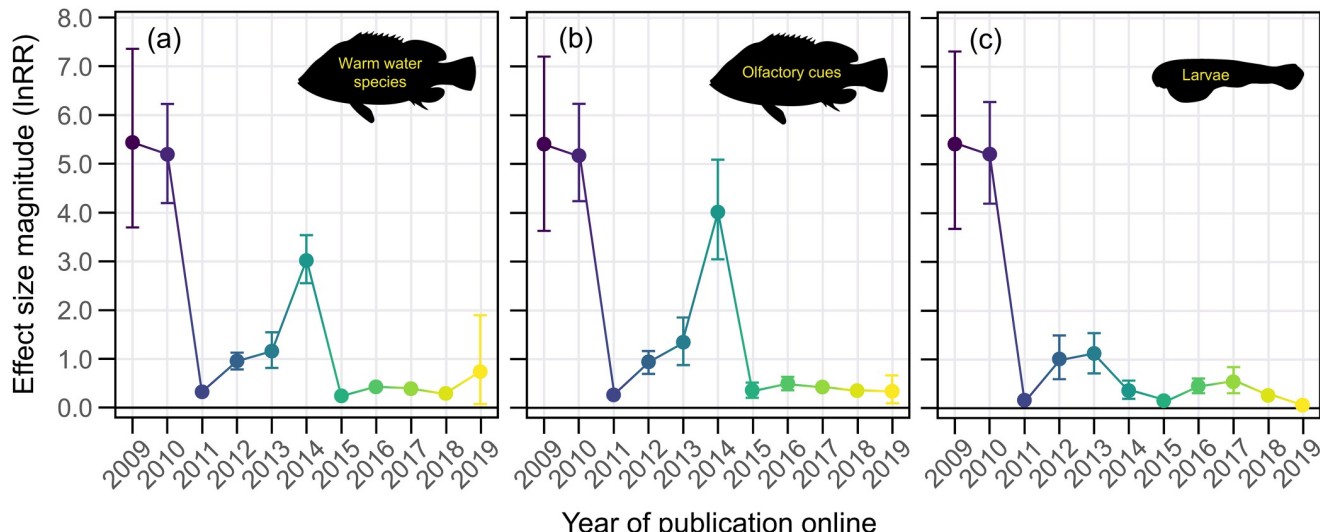

**Fig 2. The decline effect cannot be explained by 3 commonly considered biological drivers of acidification effects.** Mean effect size magnitude (absolute lnRR ± upper and lower confidence bounds) as a function of time for datasets that only included experiments with **(a)** warm-water species, **(b)** olfactory-associated behaviors, and **(c)** larval life stages. Mean effect size magnitudes and confidence bounds were estimated using Bayesian simulations and a folded normal distribution. Note: Colors are aesthetic in nature and follow a gradient according to year of publication online. Source data for each figure panel can be found in S1 Data.

comparing cold- and warm-water species, most studies do not report the actual background $CO_2$ levels that the experimental animals (and their ancestors) have historically experienced. As such, we are unable to account for the historic $CO_2$ acclimation conditions of animals used in experiments. The impact of this with respect to the observed decline effect could stem from an increasing proportion of studies using captive-bred fish from recirculating aquarium systems with high $CO_2$ levels, as compared to fish from wild populations experiencing natural $CO_2$ levels. This is an unlikely explanation for the decline effect, however, given that the earliest studies conducted in 2009 to 2010 reporting high effect sizes were conducted with both captive-bred and wild-caught fish [8–10,17]. Furthermore, recent replication attempts of those initial studies using wild-caught fish have failed to replicate the large effect sizes [7]. Nonetheless, we recommend that future studies provide better background $CO_2$ information for the fish used in their experiments and use best practices for measuring and reporting carbonate chemistry [18].

### Biased behavior in a maturing field

It is clear that the ocean acidification field, and indeed science in general, is prone to many biases including methodological and publication biases [6]. The key thing to note is that if science was operating properly from the onset, and early effects of ocean acidification on fish behavior were true, the relationships presented in Figs 1 and 2 would be flat lines showing consistent effect sizes over time. It is also evident that the decline effect discovered herein is not explainable by 3 likely biological culprits (outlined above). Thus, the data presented here provide a textbook example of a new and emerging "hot topic" field likely being prone to biases. Below, we underscore and assess the roles of 3 potential biases: (1) methodological biases; (2) selective publication bias; and (3) citation bias. We then explore the potential influence of authors/investigators in driving the decline effect.

**Methodological biases.** Methodological approaches for individual studies, and biases therein, can contribute to the early inflation of effect sizes. Such biases can come in the form of

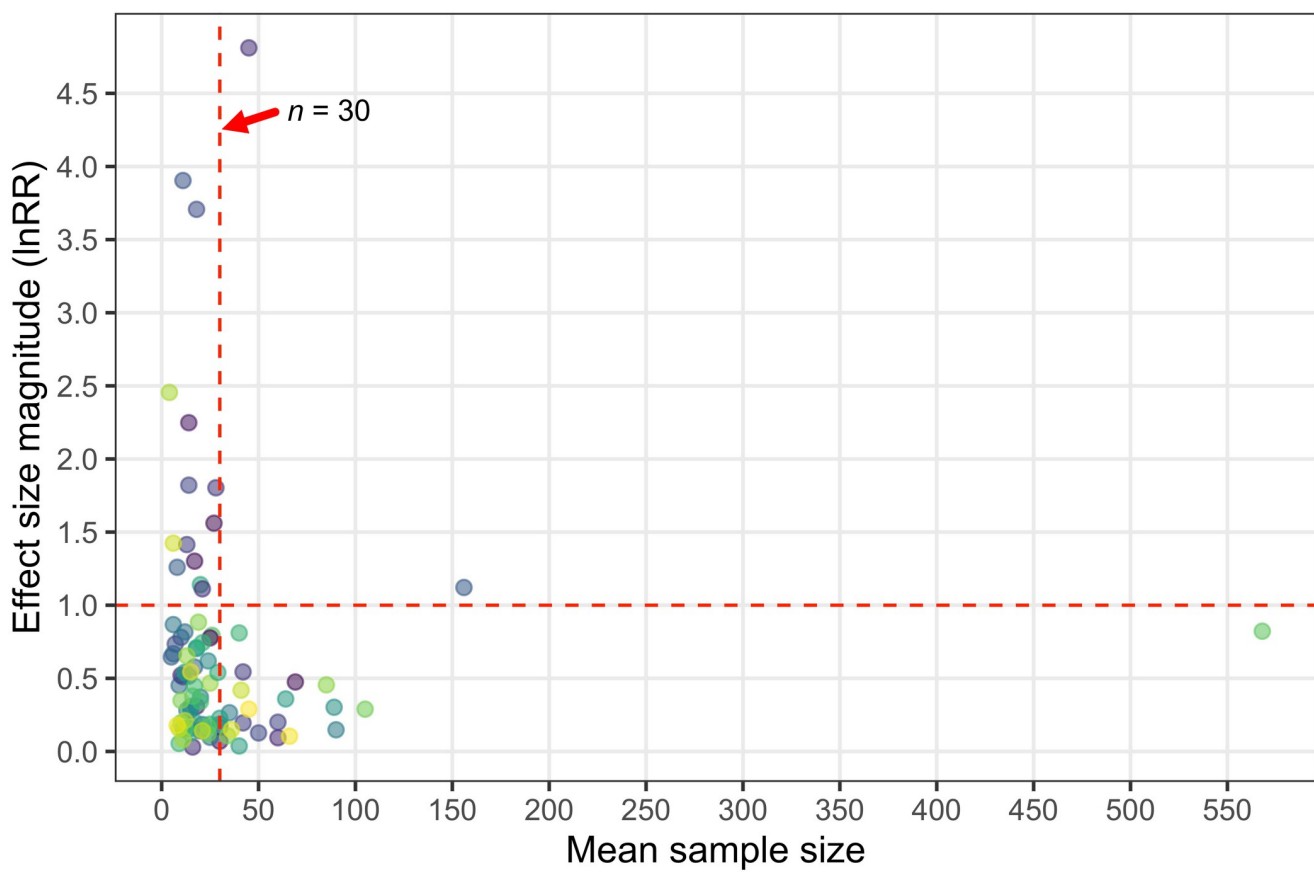

**Fig 3. Studies with large effect sizes tend to have low samples sizes.** Mean effect size magnitude (absolute lnRR) for each study as a function of the mean sample size of that study (i.e., sample size per experimental treatment). Note that mean effect size for a given study is not a weighted effect size magnitude, but is simply computed as the mean of individual effect size magnitudes for a given study. The vertical red dashed line denotes a sample size of 30 fish, while the horizontal red dashed line represents a lnRR magnitude of 1. Source data for each figure panel can be found in S1 Data.

experimental protocols, the chosen experimental design and sample size, and the analytical/ statistical approach employed. Experimenter biases can also contribute to inflated effects.

Experimental designs and protocols can introduce unwanted biases during the experiment whether or not the researchers realize it. For example, experiments with small sample sizes are more prone to statistical errors (i.e., Type I and Type II error), and studies with larger sample sizes should be trusted more than those with smaller sample sizes [19]. While we did not directly test it in our analysis, studies with small sample sizes are also more susceptible to statistical malpractices such as p-hacking and selective exclusion of data that do not conform to a predetermined experimental outcome, which can contribute to inflated effects [20]. In our analysis, we found that almost all of the studies with the largest effect size magnitudes had mean sample sizes (per experimental treatment) below 30 fish. Indeed, 87% of the studies (13 of 15 studies) with a mean effect size magnitude >1.0 had a mean sample size below 30 fish (Fig 3). Likewise, the number of studies reporting an effect size magnitude >0.5 sharply decreased after the mean sample size exceeded 30 fish (Fig 3). Sample size is of course not the only attribute that describes the quality of a study, but the effects detected here certainly suggest that studies with $n < 30$ fish per treatment may yield spurious effects and should be weighted accordingly.

Experimenter/observation bias during data collection is known to seriously skew results in behavioral research [21]. For example, nonblinded observations are common in life sciences,

but are known to result in higher reported effect sizes and more significant *p*-values than blinded observations [22]. Most publications assessing ocean acidification effects on fish behavior, including the initial studies reporting large effect sizes, do not include statements of blinding for behavioral observations. Given that statements of blinding can be misleading [23], there has also been a call for video evidence in animal behavior research [24]. Moreover, the persistence of inflated effects beyond initial studies can be perpetuated by confirmation bias, as follow-up studies attempt to confirm initial inflated effects and capitalize on the receptivity of high-profile journals to new (apparent) phenomena [25]. While our analysis does not empirically demonstrate that experimenter bias contributed to the decline effect, it is possible that conscious and unconscious experimenter biases may have contributed to large effect sizes in this field.

**Publication and citation bias.**   Another prominent explanation for the decline effect is selective publication bias, as results showing strong effects are often published more readily, and in higher-impact journals, than studies showing weak or null results. Indeed, publication bias has been suggested as perhaps the most parsimonious explanation for the decline effect in ecology and evolution, as studies showing no effect can be difficult to publish [2]. This can be attributed to authors selectively publishing impressive results in prestigious journals (and not publishing less exciting results) and also to journals—particularly high-impact journals—selectively publishing strong effects. This biased publishing can result in the proliferation of studies reporting strong effects, even though they may not be true [26] and can fuel citation bias [27]. Indeed, a recent analysis suggested that field studies in global change biology suffer from publication bias, which has fuelled the proliferation of underpowered studies reporting overestimated effect sizes [28]. To determine if studies testing for effects of ocean acidification on fish behavior exhibited signs of publication bias and citation bias, we assessed relationships between effect size magnitude, journal impact factor, and Google Scholar citations (Fig 4). Examining average citations per year and the total number of citations since 2020, 4 papers stood above the rest: the initial 3 studies in this field [8–10] and the sentinel paper proposing GABA$_A$ neurotransmitter interference as the physiological mechanism for observed behavioral effects [29] (Fig 4a and 4b). While it is difficult to quantify whether authors selectively published only their strongest effects early in this field, we were able to quantify effect size magnitudes as a function of journal impact factor. We found that the most striking effects of ocean acidification on fish behavior have been published in journals with high impact factors (Fig 4c). In addition, these studies have had a stronger influence (i.e., higher citation frequency) on this field to date than lower-impact studies with weaker effect sizes (Fig 4d and 4e). Similar results have been reported in other areas of ecology and evolution, perhaps most notably in studies regarding terrestrial plant responses to high $CO_2$ [30].

Together, our results suggest that large effect sizes among studies assessing acidification impacts on fish behavior generally have low sample sizes, but tend to be published in high-impact journals and are cited more. Consequently, the one-two punch of low sample sizes and the preference to publish large effects has seemingly led to an incorrect interpretation that ocean acidification will result in broad impacts on fish behavior and thus have wide-ranging ecological consequences—an interpretation that persists in studies published today (S2 Table).

**Investigator effects.**   It is important to note that the early studies published in 2009 to 2010 [8–10], and some subsequent papers from the same authors, have recently been questioned for their scientific validity [31]. Indeed, these early studies have a large influence on the observed decline effect in our analysis. At the request of the editors, we thus explored the potential for investigator effects, as such effects have been reported to drive decline effects for the field of ecology and evolution in the past (e.g., fluctuating asymmetry [32]). When all papers authored or coauthored by at least one of the lead investigators of those

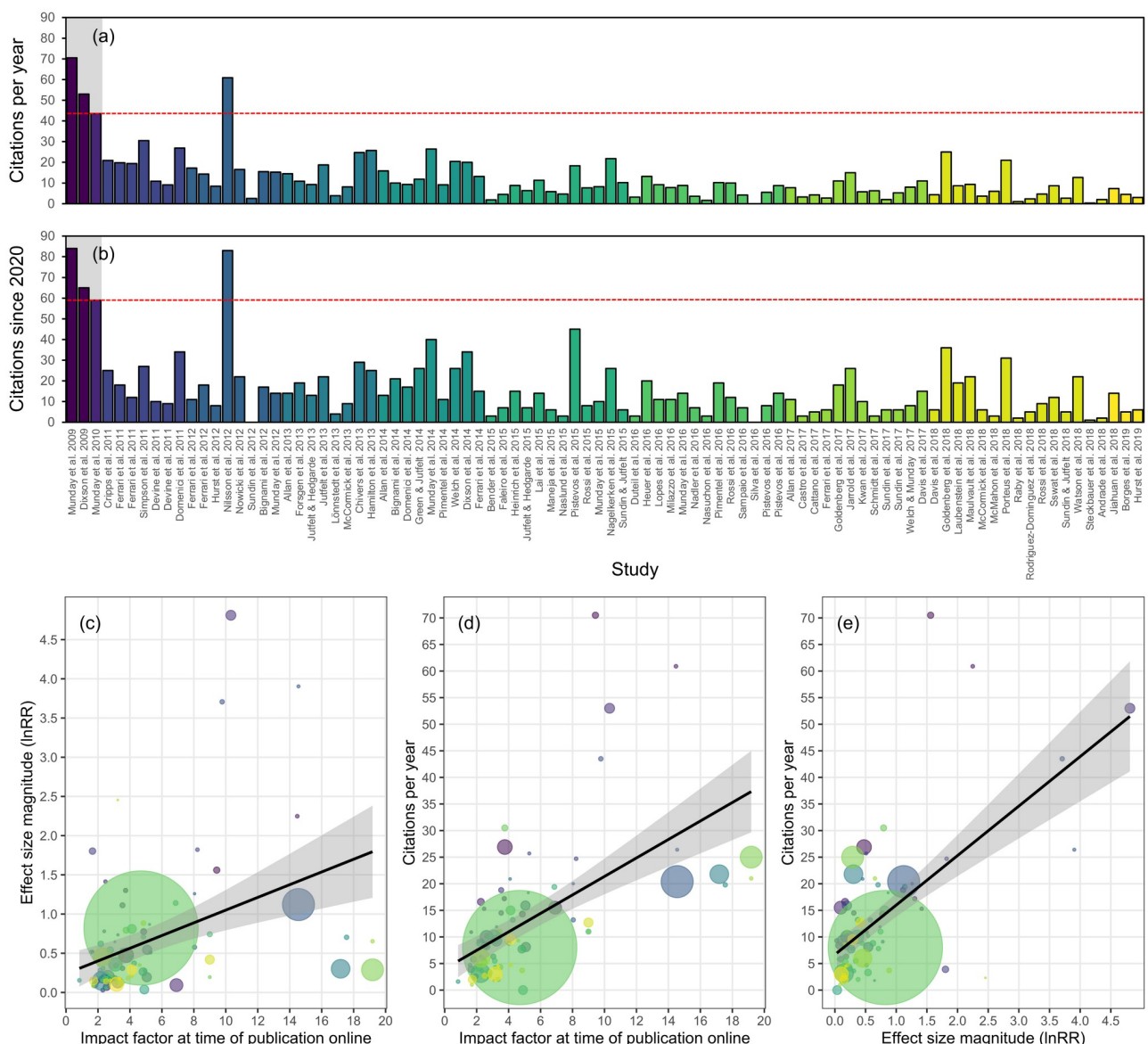

**Fig 4. Strong effects are published in high-impact journals, and these studies are cited more than small effect studies in lower-impact journals. (a, b)** Google Scholar citation metrics as of September 10, 2021 for each of the studies included in our meta-analysis, including average citations per year (a) and total citations since 2020 (b). The initial 3 studies spearheading this field are denoted by the gray background, and the red dashed line represents the lowest citation metric among those 3 studies. Studies are ordered chronologically along the x-axis and color coded by year published online. **(c)** Mean effect size magnitude for each individual study as a function of journal impact factor (at time of online publication). **(d)** The number of citations per year for each study as a function of journal impact factor (at time of online publication). **(e)** The number of citations per year for each study as a function of mean effect size magnitude for that study. Note that, for panels (c) and (e), mean effect size magnitude for a given study is not a weighted effect size magnitude, but is simply computed as the mean of individual effect size magnitudes for a given study. Data are fit with linear curves and 95% confidence bounds, and points are color coded by study; the size of data points represents the relative mean sample size of the study. Source data for each figure panel can be found in S1 Data.

early studies were removed from the dataset (*n* = 41 studies, 45%), the decline effect was no longer apparent from 2012 to 2019 (Fig 5). While conclusions regarding the potential roles of invalid data await further investigation [31], our results do suggest that investigator or lab group effects have contributed to the decline effect reported here. We suggest that future

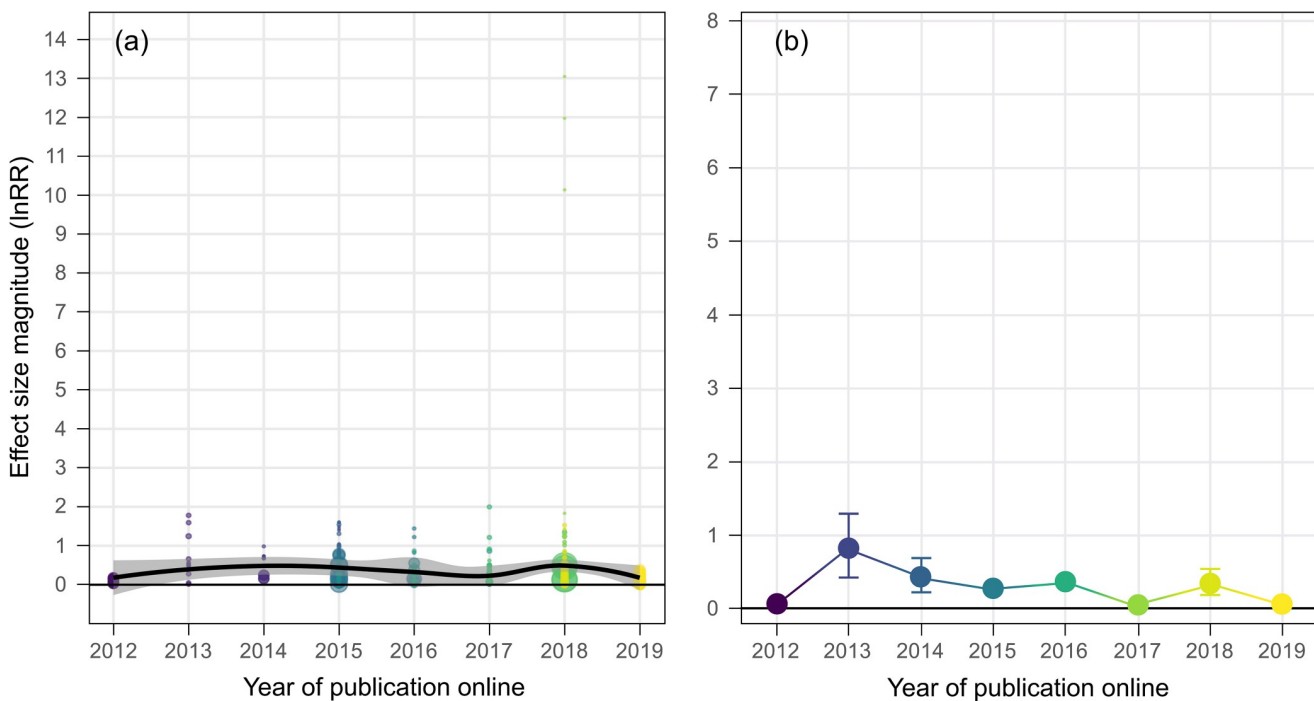

**Fig 5. The decline effect in ocean acidification research on fish behavior excluding studies authored (or coauthored) by lead investigators of initial studies.** (a) Trend in raw effect size magnitudes (absolute lnRR) for each experiment in our dataset excluding all studies authored (or coauthored) by lead investigators of the 3 initial studies [8–10] plotted as a function of year of publication online and color coded according to study. Data are fit with a Loess curve with 95% confidence bounds. (b) Mean effect size magnitude (absolute lnRR ± upper and lower confidence bounds) for each year of publication online in our dataset excluding all studies authored (or coauthored) by lead investigators of the 3 initial studies. Mean effect size magnitudes and confidence bounds were estimated using Bayesian simulations and a folded normal distribution. Note: Colors in (b) are aesthetic in nature and follow a gradient according to year of publication. Also note that data begin in 2012 since all publications prior to 2012 included initial lead investigators in the author list. Vertical axes are scaled to enable direct comparison with Fig 1. Source data for each figure panel can be found in S1 Data.

studies documenting the presence or absence of decline effects—and indeed meta-analyses in general—should carefully consider and evaluate whether investigator effects may be at play in a given field of study.

## Being on our best behavior

Our results suggest that large effects of ocean acidification on fish behavior were at least in part due to methodological factors in early studies (e.g., low sample sizes). Furthermore, the proliferation and persistence of this idea have likely been aided by the selective publication of large effect sizes by authors and journals, particularly at the onset of this field, and the continued high frequency of citations for those papers. It is important to note, however, that low sample size and selective publication cannot fully explain the strong decline effect detected here, and other biases and processes may be at play [7,31]. Nonetheless, we call on journals, journal editors, peer reviewers, and researchers to take steps to proactively address the issues of low sample size and selective publication, not only in the ocean acidification field, but also more broadly across scientific disciplines.

To this end, we strongly argue that future ocean acidification studies on fish behavior should employ a sample size greater than 30 fish per treatment in order to be considered reliable. It is the combined responsibility of researchers, journal editors, and peer reviewers to

ensure that submitted manuscripts abide by this guideline. To achieve this, authors should report exact sample sizes clearly in the text of manuscripts; however, from our analysis, 34% of studies did not do this adequately (see raw data in S2 Data). In addition, for other fields, we suggest that studies with higher sample sizes should be published alongside, if not very soon after, an original novel finding to ensure that such a finding is robust. Ideally, researchers would conduct pilot studies with varying sample sizes to determine an adequate sample size threshold and conduct appropriate prestudy power analyses; however, time and financial constraints can make this difficult. While adequate sample sizes will vary across topics and fields, ensuring that studies with large sample sizes are published early alongside those with smaller sample sizes can strive toward reducing the amount of time it takes to truly understand a phenomenon.

Journals, researchers, editors, and reviewers can take additional steps to limit biases in published research. First and foremost, we suggest that journals adopt the practice of registered reports to ensure that studies not detecting an effect are published in a timely manner. Herein, journals should provide authors with the ability to submit proposed methodologies and have them formally peer reviewed prior to studies even being conducted. If methodologies are deemed sound (or revised to be so) and "accepted" by reviewers, journals should commit to publishing the results regardless of their outcome so long as the accepted methods are followed. Although registered reports may not be sufficient to avoid the influence of some issues such as poor data, they may reduce the risk of inflated results driving decline effects—and prolonged incorrect understanding—for other phenomena in the future. While not a silver bullet solution, this practice could help to reduce selective publication bias and the risk of early, flawed studies being disproportionately influential in a given field [33].

Researchers should also seek, develop, and adhere to best practice guidelines for experimental setups [34] to minimize the potential for experimental artifacts to influence results. Properly blinded observations [22] and the use of technologies such as automated tracking [35] and biosensors [36] can also reduce observer bias and increase trust in reported findings [37]. When automated methods are not possible, video recordings of experiments from start to finish can greatly increase transparency [24]. Editors and the selected peer reviewers should closely consider and evaluate the relevance and rigor of methodological approaches, which can help increase accuracy and repeatability [38]. When selecting peer reviewers for manuscripts, editors should also be aware that researchers publishing initial strong effects may be biased in their reviews (i.e., selectively accepting manuscripts that support their earlier publications) and ensure a diverse body of reviewers for any given manuscript when possible. While we do not empirically demonstrate this bias in our analyses, it is important to recognize and mitigate the potential for it to prolong inaccurate scientific findings.

Finally, being critical and skeptical of early findings with large effects can help avoid many of the real-world problems associated with inflated effects. Interestingly, a recent study showed that experienced scientists are highly accurate at predicting which studies will stand up to independent replication versus those that will not [39], lending support to the idea that if something seems too good to be true, then it probably is. Nonetheless, the citation analysis provided herein suggests that researchers have been slow to adopt studies reporting negative and null results for this field, as the early studies with large effect sizes remain the most highly cited among all articles in our dataset. The earlier that a healthy skepticism is applied, the less impact inflated results may have on the scientific process and the public perception of scientists. Ultimately, independent replication should be established before new results are to be trusted and promoted broadly.

## Final remarks

Our results demonstrate that more than a decade of ocean acidification research on fish behavior is characterized by the decline effect. While the field has seemingly settled in a good place with respect to realistic effect sizes, it has taken 10 years to get there. Furthermore, studies continue to cite early studies with unreasonable effect sizes to promote that acidification will broadly impact fish behavior and ecology (e.g., S2 Table), suggesting that a shift in mindset is still needed for many in this field. In a broader sense, our data reveal that the decline effect warrants exploration with respect to other biological and ecological phenomena and a wider array of scientific disciplines, particularly pertaining to global change effects. The early exaggeration of effects can have real impacts on the process of science and the scientists themselves [40]; following the steps outlined here can help to mitigate those impacts, sooner get to a real understanding of a phenomenon, and progress toward increased reproducibility.

## Materials and methods

### Literature search

A systematic literature search was conducted according to the Preferred Reporting Items for Systematic Reviews and Meta-Analyses (PRISMA) guidelines [41]; a completed PRISMA checklist can be found in S3 Data, and a flowchart is provided below (Fig 6). Peer-reviewed articles assessing the effects of ocean acidification on fish behavior were searched for through Scopus and Google Scholar by J.C. Clements up until December 21, 2018 using 2 primary keyword strings: "ocean acidification fish behavio(u)r" and "elevated co2 fish behavio(u)r." The search was conducted using the free software "Publish or Perish" [42] selecting a time period spanning 2009 to 2018 and the maximum number of results that the software allows (1,000 results), ignoring citations and patents. The keyword search resulted in a total of 4,411 results, with 2,508 papers remaining for initial screening after duplicates were removed (Fig 6, S3 Table). The titles and abstracts of each article were then screened for initial relevance and inclusion criteria. Articles were included in the database if they included statements of quantitatively assessing the effect of elevated $CO_2$ (i.e., ocean acidification) on a behavioral trait of a marine fish; we excluded review articles and papers that measured the effect of elevated $CO_2$ on freshwater fish and invertebrates. This initial screening resulted in a total of 93 papers being retained from the database search for further evaluation. Five papers were subsequently excluded from the meta-analysis due to a lack of appropriate data for estimating effect size (i.e., variance and/or sample sizes were not a part of the behavioral metric, or specific behavioral data were not presented), resulting in a total of 88 papers. A cited reference search of the 93 articles was subsequently conducted by J.C. Clements on March 23, 2019 (just prior to conducting the data analysis) by searching the reference lists and lists of citing articles (on the article's web page), selecting articles with relevant titles, and evaluating them for inclusion according to the criteria above. Three additional relevant papers were added from the cited reference search for a total of 91 papers included in the meta-analysis. While we did not solicit a call for gray literature, which can be important for meta-analyses [3], such literature online would have been captured in the Google Scholar search; however, no relevant gray literature was uncovered in this search. Final checks of the 91 papers were conducted by both J.C. Clements and J. Sundin. Results of the literature search are provided in Fig 6 below. Further details can be found in S3 Table, and full search results for each step can be found in S4 Data.

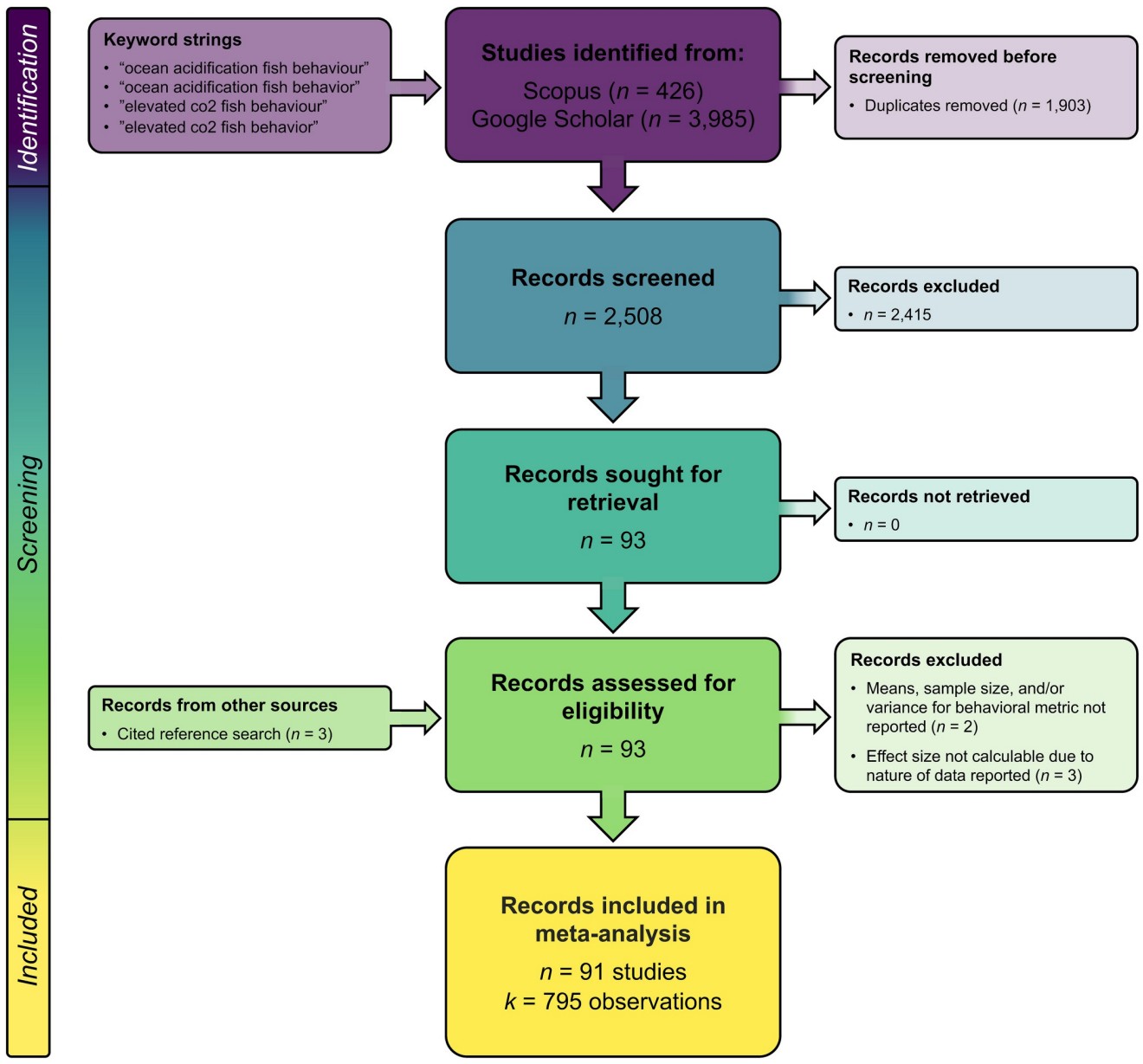

**Fig 6. PRISMA flow diagram.** Values represent the numbers of records found and retained at each stage of the literature search. Papers were considered "relevant" if they included an empirical test of ocean acidification on the behavior of a marine fish. Off-topic papers and topical review papers were excluded, as were topical papers on freshwater species and invertebrates. Relevant studies were deemed "ineligible" if they did not contain data from which effect sizes could be calculated (this included data that did not have an associated sample size or variance or relevant papers that did not report the behavioral data). Details of relevance and exclusion can be found in S4 Data. PRISMA, Preferred Reporting Items for Systematic Reviews and Meta-Analyses.

## Data collection

We collected both qualitative and quantitative data from each study. All raw data (both qualitative and quantitative) can be found in S2 Data.

**Qualitative data collection.** From each of the 91 articles, we collected general bibliographic data, including authors, publication year, title, journal, and journal impact factor. For publication year, we recorded the year that the article was published online as well as the year

that the article was included in a printed issue. Journal impact factor was recorded for the year of publication as well as the most current year at the time of analysis (2017); papers published in 2018 and 2019 were assigned to the impact factor for 2017 since 2018 and 2019 data on impact factor were unavailable at the time of analysis. Impact factors were obtained from InCites Journal Citation Reports (Clarivate Analytics). We also recorded other qualitative attributes for each study, including the species and life stage studied, and the behavioral metric(s) measured.

**Quantitative data collection.** Alongside qualitative data, we also collected quantitative data from each of the 91 studies included in the meta-analysis. We collected the mean, sample size, and variance associated with control and ocean acidification treatments. We considered all ocean acidification treatments in our analysis; however, we only included data for independent main effects of ocean acidification, and interactive effects of acidification with other factors (temperature, salinity, pollution, noise, gabazine, etc.) were ignored.

Where possible, precise means and variance were collected from published tables or published raw data; otherwise, means and variance were estimated from published graphs using ImageJ 1.x [43]. Sample sizes were obtained from tables or the text or were backcalculated using degrees of freedom reported in the statistical results. We also recorded the type of variance reported and, where possible, used that to calculate standard deviation, which was necessary for effect size calculations. Again, these data were not obtainable from 5 papers, due to either the nature of the data (i.e., no variance associated with the response variable measured or directional response variables measured in degrees; the latter due to computational issues arising from such metrics) [44–46] or from the paper reporting an effect of ocean acidification but not adequately providing the means and/or variance in neither the paper or supplementary material [47,48]. Where means and variance were measurable but observed to be zero, we estimated both as 0.0001 in order to calculate effect size [8–10,17,45,49–53]. The data were used to generate effect sizes and variance estimates for each observation. All data were initially collected by J.C. Clements and cross-checked by coauthors for accuracy prior to analyses.

## Meta-analysis

**Testing for the decline effect.** To assess whether or not a decline effect was evident in ocean acidification research on fish behavior, we used 2 approaches: (1) visualizing the trend of raw effect size magnitudes for all experiments in the dataset over time; and (2) computing weighted mean effect size magnitudes for each year in our dataset and assessing the trend in mean effect size magnitudes over time.

**Visualizing the decline effect using raw effect size magnitudes.** First, we computed the raw effect size magnitude for each individual observation in our dataset and simply visualized the trend in these effect sizes over time (i.e., Fig 2a). The effect size of choice was the natural logarithmic transformed response ratio, lnRR, which is calculated as

$$lnRR = ln\left(\frac{\bar{X}_E}{\bar{X}_C}\right),$$

where $\bar{X}_E$ and $\bar{X}_C$ are the average measured response in the experimental and control treatments, respectively. We chose lnRR because it is commonly used in ocean acidification research [54–57] and is appropriate for both continuous and ratio type response variable data (i.e., proportions and percentages, which were abundant in our dataset) that are commonly used in behavioral studies [58,59]. Using lnRR does have drawbacks, however. Mainly, lnRR cannot be calculated when a response variable has a positive value for one treatment group and a negative value for the other. As such, we excluded measures of relative behavioral lateralization (a measure of left–right turning preference) from our analysis, as well as any index metrics that spanned positive and negative values. For response variables that were reported as a

"change in" behavior from a specific baseline (and could therefore have both positive and negative values), we only included instances in which the response variable values for the control treatment and elevated $CO_2$ treatment were both of the same directionality (i.e., both positive or both negative changes). For all such instances, the rationale for omissions and/or inclusion are provided in the "Notes" column in S2 Data.

Once calculated, the individual effect sizes were transformed to the absolute value due to the inherent difficulty in assigning a functional direction to a change in behavior, as many behavioral changes can be characterized by both positive and negative functional trade-offs. For example, increased activity under elevated $pCO_2$ could make prey fish more difficult for predators to capture, but could also make prey more noticeable to predators. Therefore, rather than prescribing arbitrary functional directionality to altered behavior, we simply elected to use absolute value (i.e., unsigned value) of lnRR to visualize the decline effect. It is important to note that such a transformation only provides a measure of effect size magnitude. Thus, the absolute effect size overestimates and is therefore a conservative estimate of the true effect size, but can still be used to test for declining effect size magnitudes over time (and can thus be used to test for the decline effect). Although this can complicate true population-level inferences [60], the use of absolute effect size values is informative for understanding the strength of effects ignoring directionality [61].

**Assessing weighted mean effect size magnitudes by year.**    Although useful for visualizing a trend in effect sizes over time, the first approach above is not analytically rigorous. Properly analyzing trends in effect sizes should including a weighted component whereby individual effect sizes are weighted according to their precision (i.e., measurements with a larger sample size and lower variance should be given more weight than those with a lower sample size and higher variance) [62]. As such, we computed weighted mean effect size magnitudes (and their associated uncertainty, i.e., upper and lower confidence bounds) for each year represented in our dataset and assessed the trend in these effect sizes over time.

Weighted mean effect size magnitudes (lnRR) and their confidence bounds were computed using the "transform-then-analyze" approach as suggested by [63], with R code adapted from [64] to avoid biased estimates of effect size magnitude. Briefly, this method estimates the mean effect size for each level of a moderator of interest (i.e., each year in our dataset) by assuming a normal distribution and subsequently transforming the mean effect size using a folded normal distribution to estimate a mean effect size magnitude. Uncertainty around the mean effect size magnitude was estimated in a Bayesian fashion using the MCMCglmm() function from the *MCMCglmm* package [65], applying the entire posterior distribution of mean estimated to the folded normal distribution as per [64]. For analytical reproducibility, the supporting information includes annotated R code (S1 Code), source data for each figure panel (S1 Data), and raw data files used for analysis (S5 to S13 Data).

## Assessing biological explanations for the decline effect

Since a decline effect was detected in our analysis, we explored 3 biological factors that might explain the observed decline effect: (1) climate (cold-water versus warm-water species); (2) behavior type (olfactory versus nonolfactory behaviors); and (3) life stage (larvae versus juveniles and adults).

Because early studies were focused on warm-water fish from tropical coral reefs, the observed decline effect could potentially be driven by an increasing number of studies on less sensitive cold-water species over time. Cold-water fish in temperate regions experience a higher degree of temporal variability in carbonate chemistry parameters over large spatial areas [15]. Therefore, they may be less sensitive to changes in seawater $CO_2$ as per the Ocean

Variability Hypothesis [16]. If an increasing number of studies on cold-water species over time was responsible for the decline effect, removing cold-water species from the dataset (i.e., only including warm-water species) should result in the decline effect trend disappearing. In the same vein, the strongest effects of ocean acidification on fish behavior have undoubtedly been reported for olfactory responses, and an increasing number of studies on nonolfactory behaviors could explain the decline effect. If this was true, removing nonolfactory behaviors from the dataset should negate the decline effect trend. We therefore tested for the influence of non-olfactory behaviors by removing them from the dataset and rerunning the analysis. Finally, larvae are typically considered to be more sensitive to acidification than juveniles and adults, and removing less sensitive life stages from the dataset would remove the decline effect trend if this explanation was responsible for the decline (i.e., if studies using less sensitive life stages had increased proportionally over time). Therefore, to test whether or not the decline effect was due to these 3 biological factors, we reran the analysis described in the **Assessing weighted mean effect size magnitudes by year** section above on 3 separate datasets: one with cold-water species removed, one with nonolfactory responses removed, and one with juvenile and adult life stages removed.

## Assessing evidence for selective publication bias, citation bias, methodological bias, and investigator effects

Alongside testing for the decline effect, we also wanted to determine whether publication bias and/or methodological bias may have contributed to the large effect sizes reported in this field and whether there was any evidence for citation bias. In new and emerging topics, large effect sizes can be driven by authors and high-impact journals selectively publishing novel and groundbreaking results with large effect sizes [66]. If selective publication bias was evident among studies testing for effects of ocean acidification on fish behavior, there would be a positive relationship between effect size magnitude and journal impact factor *sensu* [30]. Thus, to determine if selective publication bias could be present in this field, we visually assessed the relationship between the journal impact factor (for the year of online publication) and the mean effect size magnitude for each study. It is important to note here that we did not compute weighted mean effect size magnitudes for each study, but simply computed the mean of the raw effect size magnitudes as calculated in the section **Visualizing the decline effect using raw effect size magnitudes** above.

To check for citation bias, we visually assessed the relationship between impact factor and the number of citations per year (according to Google Scholar on September 10, 2021) for each study, as well as the relationship between mean effect size magnitude and citations per year. We chose to use Google Scholar for citation data because Scholar has been shown to be more comprehensive than other sources (e.g., Web of Science, Journal Citation Reports, and Scopus), as it not only captures the vast majority of citations documented by these other sources, but also tends to capture more citations that are missed by those sources [67,68]. If citation bias was present in this field, citations per year would be positively correlated with mean effect size magnitude. Furthermore, if selective publication bias was influencing citation bias, a positive relationship between impact factor and citations per year would be present.

To assess if low sample sizes could contribute to large effect sizes (i.e., higher probability of Type 1 error), we plotted mean effect size magnitude for each study against the mean sample size of that study. If low sample size was influencing effect sizes among studies in this field, large effect sizes would cluster near the lower end of the sample size spectrum.

Finally, because the validity of data presented in the early studies of this field have recently been questioned [31], and investigator bias has been reported to drive decline effects in ecology

and evolution in the past [32], we were asked by the editors to test for investigator (or lab group) effects by rerunning the analysis on a dataset with all papers authored or coauthored by the lead investigators of those initial papers (i.e., P. Munday and/or D. Dixson) removed. Herein, we once again visualized all raw effect sizes plotted against time (i.e., see **Visualizing the decline effect using raw effect size magnitudes** section) and also computed weighted mean effect size magnitudes for each year (i.e., see **Assessing weighted mean effect size magnitudes by year** section). The potential for investigator effects influencing the decline effect would be apparent if the decline effect was not evident in the dataset excluding these authors.

## Supporting information

**S1 Code. R code.** Annotated R code for analysis and figure generation.
(R)

**S1 Data. Figure data.** Underlying numerical data for individual figure panels, including Figs 1a, 1b, 2a, 2b, 2c, 3, 4a, 4b, 4c, 4d, 4e, 5a and 5b.
(XLSX)

**S2 Data. Raw data.** Complete dataset including meta-data, study (and observation) specific qualitative and quantitative data, and information on excluded studies.
(XLSX)

**S3 Data. PRISMA checklists.** Completed PRISMA 2020 checklists. PRISMA, Preferred Reporting Items for Systematic Reviews and Meta-Analyses.
(DOCX)

**S4 Data. Literature search results.** Detailed breakdown of the systematic literature search results.
(XLSX)

**S5 Data. Raw data file.** Data file for analysis in R. This dataset applies to the analysis titled "##### META-ANALYSIS–YEAR ONLINE–FULL DATASET" in S1 Code.
(CSV)

**S6 Data. Raw data file.** Data file for analysis in R. This dataset applies to the analysis titled "##### META-ANALYSIS–YEAR ONLINE–WARM WATER ONLY" *in* S1 Code.
(CSV)

**S7 Data. Raw data file.** Data file for analysis in R. This dataset applies to the analysis titled "##### META-ANALYSIS–YEAR ONLINE–OLFACTORY CUES ONLY" in S1 Code.
(CSV)

**S8 Data. Raw data file.** Data file for analysis in R. This dataset applies to the analysis titled "##### META-ANALYSIS–YEAR ONLINE–LARVAE ONLY" in S1 Code.
(CSV)

**S9 Data. Raw data file.** Data file for analysis in R. This dataset applies to the analysis titled "##### META-ANALYSIS–INVESTIGATOR EFFECTS" in S1 Code.
(CSV)

**S10 Data. Raw data file.** Data file for analysis in R. This dataset applies to the analysis titled "##### CREATE SCATTERPLOT FIGURE TO VISUALIZE MEAN EFFECT SIZE MAGNITUDE FOR EACH OBSERVATION OVER TIME (Fig 1A)" in S1 Code.
(CSV)

**S11 Data. Raw data file.** Data file for analysis in R. This dataset applies to the analysis titled "##### CREATE SCATTERPLOT FIGURE TO SAMPLE SIZE BIAS (Fig 3)" in S1 Code.
(CSV)

**S12 Data. Raw data file.** Data file for analysis in R. This dataset applies to the analysis titled "##### CREATE SCATTERPLOT FIGURE TO VISUALIZE PUBLICATION BIAS (Fig 4C–4E)" in S1 Code.
(CSV)

**S13 Data. Raw data file.** Data file for analysis in R. This dataset applies to the analysis titled "##### CREATE SCATTERPLOT FIGURE TO VISUALIZE INVESTIGATOR EFFECTS (Fig 5)" in S1 Code.
(CSV)

**S1 Table. Mean effect size magnitudes and their uncertainty depicting the decline effect.** Mean effect size magnitudes and their upper and lower confidence bounds for each dataset. Mean effect sizes were estimated by assuming a normal distribution and subsequently transforming the mean effect size using a folded normal distribution. Uncertainty around the mean effect size magnitude was estimated in a Bayesian fashion (see Materials and methods). mean mag = mean effect size magnitude; CI LB = lower confidence bound; CI UB = upper confidence bound.
(DOCX)

**S2 Table. Studies continue to reference early studies to state that ocean acidification is predicted to have wide-ranging effects on fish behavior and ecology.** Selected quotes pulled from 4 papers published in 2021 stating that ocean acidification is predicted to have broad impacts on fish behavior.
(DOCX)

**S3 Table. Literature search results.** Expanded details for each stage of the literature search, including results for each keyword and each database. Full search results can be accessed in S4 Data.
(DOCX)

## Acknowledgments

We thank Christophe Pélabon (Norwegian University of Science and Technology) for statistical advice and many discussions surrounding this project at the onset of the study. We also thank Dr. Daniel Noble (Australian National University) and Dr. Alfredo Sánchez-Tójar (Bielefeld University) for further statistical advice for analyzing effect size magnitudes. Thanks also to Dr. Steven Novella (Skeptics Guide to the Universe podcast) and Neuroskeptic for the inspiration to investigate the decline effect in this field.

## Author Contributions

**Conceptualization:** Jeff C. Clements, Fredrik Jutfelt.

**Data curation:** Jeff C. Clements, Josefin Sundin.

**Formal analysis:** Jeff C. Clements.

**Funding acquisition:** Jeff C. Clements, Fredrik Jutfelt.

**Investigation:** Jeff C. Clements, Josefin Sundin, Timothy D. Clark, Fredrik Jutfelt.

**Methodology:** Jeff C. Clements, Josefin Sundin, Timothy D. Clark, Fredrik Jutfelt.

**Project administration:** Jeff C. Clements, Fredrik Jutfelt.

**Resources:** Jeff C. Clements, Timothy D. Clark, Fredrik Jutfelt.

**Software:** Jeff C. Clements.

**Supervision:** Jeff C. Clements, Fredrik Jutfelt.

**Validation:** Jeff C. Clements, Josefin Sundin, Timothy D. Clark, Fredrik Jutfelt.

**Visualization:** Jeff C. Clements.

**Writing – original draft:** Jeff C. Clements.

**Writing – review & editing:** Jeff C. Clements, Josefin Sundin, Timothy D. Clark, Fredrik Jutfelt.

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
