## [Editor Report · Decision Letter 0]

11 Jan 2021

Dear Dr Clements, 

Thank you for submitting your manuscript entitled "An extreme decline effect in ocean acidification fish ecology" for consideration as a Research Article by PLOS Biology. Please accept my apologies for the unusual delay; you submitted at a time when we had closed our journal office for two weeks over the holiday period.

Your manuscript has now been evaluated by the PLOS Biology editorial staff, as well as by an academic editor with relevant expertise, and I'm writing to let you know that we would like to send your submission out for external peer review.

IMPORTANT: We think that your paper would be better considered as a Meta-Research Article; these very widely read articles deal with questions about how research is performed and reported, and you can see previous examples by running this search: https://journals.plos.org/plosbiology/search?q=article_type%3A%22meta-research+article%22&page=1 There is no need for any reformatting, but please select "Meta-Research Article" as the article type when you upload your additional meta-data (see next paragraph).

Please re-submit your manuscript within two working days, i.e. by Jan 13 2021 11:59PM.

Kind regards,

Roli Roberts

Senior Editor

PLOS Biology

---

## [Decision Letter · Decision Letter 1]

15 Mar 2021

Dear Dr Clements,

Thank you very much for submitting your manuscript "An extreme decline effect in ocean acidification fish ecology" for consideration as a Meta-Research Article at PLOS Biology. Your manuscript has been evaluated by the PLOS Biology editors, an Academic Editor with relevant expertise, and by four independent reviewers.

You'll see that while some of the reviewers are broadly positive about your study, there are also some substantial concerns raised (especially by reviewers #1 and #4), and these must be addressed for further consideration. Broadly speaking, reviewers #1 and #3 work on marine ecology and fish behaviour, including the effects of acidification, and reviewers #2 and #4 are experts in meta-analysis (with an ecology background). Reviewer #1 is concerned about your treatment of two early outlier studies, both reviewer #2 and #4 ask that you use PRISMA reporting, and reviewer #4 has a number of other significant methodological concerns. Reviewer #3's requests are largely textual.

In light of the reviews (below), we will not be able to accept the current version of the manuscript, but we would welcome re-submission of a much-revised version that takes into account the reviewers' comments. We cannot make any decision about publication until we have seen the revised manuscript and your response to the reviewers' comments. Your revised manuscript is also likely to be sent for further evaluation by the reviewers.

We expect to receive your revised manuscript within 3 months. 

**IMPORTANT - SUBMITTING YOUR REVISION**

*Re-submission Checklist*

*Published Peer Review*

*PLOS Data Policy*

*Blot and Gel Data Policy*

Sincerely,

Roli Roberts

Senior Editor,

rroberts@plos.org,

PLOS Biology

REVIEWERS' COMMENTS:

Reviewer #1:

Summary: The current paper seeks to explore the possibility that ocean acidification research related to behavioral effects on fishes may suffer from a decline effect, and to also explore possible sources of this potential effect. They utilize a combination of qualitative and quantitative meta-analysis approaches to demonstrate (in their view) a decline effect, and to suggest that this effect is not due to biological processes. They conclude that ocean acidification does not have as a great an effect as previously thought. While I can see the rationale for this study, I honestly was not convinced by the work (for reasons I will outline) and was underwhelmed by their attempts to explain any observed effect, with the exception of the sample size discussion. There are a number of reasons, biological, methodological, and external, that may explain some of the observed shifts in effect size. 

Major Comments: 

#1) When looking at the data, it is clear that this entire trend is built on the presence of two early papers published on the topic, Munday et al 2010 and Dixson et al. 2010. I will obviously agree that the effect sizes in these papers are larger than typically found elsewhere in the literature, and this is not argued in the community. These papers were published a decade ago and to say that they are held up as representative of the field would be grossly inaccurate. But I have two other criticisms of the analysis with regard to these papers. First is that Dixson et al. 2010 appears to be classified under the 2009 data, which causes the 2009 reported value to be inflated in support of the decline effect. It might actually be included with both the 2009 and 2010 data. Confidence intervals are provided for both years, but one of the years should have only a single nested mean data point. A confidence interval can't be created around a single data. Anyway, I disagree with the use of the online date instead of official citation date (seriously, it was published online in late November). Dixson et al should be placed with the 2010 data, so the mean 2009 value would be 1.41 (the nested mean of Munday et al 2009, the only paper published that year). This all but destroys the decline effect trend as reported. This shows how sensitive the authors conclusions are to arbitrary choices made during data analysis. 

My second point about these two papers (this actually applies throughout) is the practice of nesting the effect sizes to generate a mean effect size per paper. Why is a mean valuable here? The data are clearly not normally distributed within a paper (nor is there any reason to believe they would be), so a mean is a poor representation of the combined data. This might be what the authors are referring to in the Figure 1 caption, but then they proceed to base their conclusions off of invalid overestimated means. If you calculate the median for these two papers they are 0.58 and 1.07, which are comparable to median values in later years. My argument is not to use the median, but in fact to use all the raw data. There is no need to reduce the data set with nesting, at least when attempting to initially identify a decline effect. When I generated a scatterplot of all the effect sizes by year, the decline trend is not apparent. There are effect sizes in excess of 13 as late as 2018. In other words, the identified decline effect is a product of selective data analysis by the authors, the trend is not robust to varying forms of analysis, and the trend is largely driven by two outlier papers published in the first two years. 

#2) The authors claim the observed trend (assuming it's true) cannot be explained biologically because they performed a cold water vs warm water comparison. The basis for this comparison is not explained, and it is not clear to me why cold water fish behavior would be less impacted by ocean acidification. This needs to be clearly articulated to the reader because to my understanding this comparison is not rooted in any sort of underlying physiology. Ocean acidification effects have nothing to do with metabolic rate. 

#3) Building on comment #2, Baumann 2019 (CJZ) has proposed that coastal species that experience high CO2 at various points in their life cycle are more resilient to its effects, and there is support for this line of thinking (i.e. acclimation and transgenerational plasticity). This seems like a much better biological test than a spurious latitudinal comparison. Furthermore, the authors must also address the increased use of captive bred fish populations. High CO2 is a very well-known problem in aquaculture, and fish reared in captivity could be acclimated to such conditions, or even selected to perform well in such conditions. 

#4) The authors propose that selective publication bias may be, in part, to blame for the decline effect. I would argue the exact opposite. Ocean acidification is unique in environmental science as a topic in which you can readily publish a completely negative results data set (in Nature no less). I would challenge the authors to find any other environmental stressor where this publication access is available to scientists. I am not arguing it is a bad thing, negative results are valuable in the context of climate change, but the authors ignore the fact that the field has been littered with low impact journal no-effect papers. In fact, the authors cite a paper by Browman (2016 ICES Marine Science) that served as a "call-to-arms" to publish negative results papers and ICES is an excellent place to do it. So my question is this: is the observed decline effect because of an atypical decrease in effect size, or is it because of an atypical number of no-effect publications that would not have been publishable under different circumstances. For example, is a no-effect CO2 paper on fish behavior publishable in 2006? Probably not. Again, I would argue the authors are not viewing their data set impartially. The field seems to have settled in a good place, behavior effects are not pervasive but may be a part of the overall ocean acidification story. Isn't this the way the system is supposed to work? 

#5) I see no benefit to the qualitative analysis. You have quantitative data to rely on. 

#6) I would have liked to see some attempt to divide the data based on behavior type. The authors did try to address this a little bit with the baseline vs cue data, but I would argue that it really isn't mechanistically valid to combine olfactory endpoints, lateralization endpoints and anxiety type endpoints in a single analysis. Or at the very least, they should also be analyzed separately. 

#7) This wasn't mentioned by the authors, but I've always wondered whether the shift in sensitivity had anything to do with differences in the way CO2 was calculated. Munday et al 2010 used a submerged gas probe with a permeable membrane, as opposed to the many studies that quantify pH and titratable alkalinity. Dixson only relied on pH measurements (no alkalinity) and doesn't delineate what type of pH scale it is. The real PCO2 could be off by quite a bit from what is reported. That method would never be published today. If the authors are actually interested in looking at all possible explanations for this outlier paper (and I think their decline effect is more an outlier paper effect), I think it is reasonable to include the analytical chemistry as an explanation. 

#8) Throughout the manuscript the authors place a major emphasis on the effect size, presumably because greater effect sizes are related to more detrimental impacts of ocean acidification. But the authors should be careful to not equate these things too directly. For example, just say instead of impacting 100% of individuals, OA will only impact 30%. Is that ecologically meaningless? In 2018 the effect size 95% CI does not overlap with zero, so there is still an effect there. The ecological significance of the effect is the question. 

Specific Comments: 

Line 16: "is expected to". This wording is misleading of the field. Ocean acidification has been intensely studied and its effects are varied. I would argue that the general community views ocean acidification as much less of a threat than a decade ago, but that there are specific circumstances where it may cause problems. 

Line 18: "most catastrophic". Again, this is a misrepresentation. The people that study behavior (like the authors) place it amongst the most catastrophic effects, but I'm not sure the entire community agrees. I would suggest to tone down the language here. 

Line 19: "dire prediction". The authors are framing this manuscript as if it is 2011. These dire predictions are not roundly held by the community. There are a number of review papers that have challenged the GABA hypothesis, and pointed out that the effects of OA on behavior are not widely observed amongst fishes, at least at OA relevant CO2 levels. 

Line 23: You tried one thing. This is a gross overstatement. 

Line 25: This conclusion is again rooted in 2011. There is unanimous understanding that the effects of OA on behavior are not universal, and as such the early "dire" predictions were abandoned year ago. That isn't to say that there are not behavior based effects that can have serious consequences. I would tone this down to be more representative of the field. 

Line 29: I was a little surprised that a paper on OA effects didn't really introduce the topic in a meaningful way. 

Line 37: Rephrase. Inflated initial reports infers some sort of deliberate intent, which I don't think is what the authors are trying to say. Also applies to the next sentence. 

Line 47: Again, this subheading infers deliberate intent. 

Line 48: I would argue that very few have documented negative effects. That requires a very high power, which most studies do not have. The authors cannot argue for the importance of sample size for type 1 error but ignore it for type 2 error. In reality, most of the no effects papers are unable to demonstrate a positive effect but inconclusive with regard to no effect. 

Line 50: I would say temperature is a pretty nice precedent. Or maybe the effects of oil following the Exxon oil spill, or Deepwater Horizon. Maybe the current explosion of PFAS work. I know PLOS Biology is a high impact journal, but the hyperbole in this manuscript is over the top. 

Line 54: outstanding to large? 

Line 62: I'm pretty sure the universality of behavioral effects was abandoned long ago. 

Line 65: This might be a good place to actually expand on the types of behaviors. 

Line 99: How is an increased study of cold water species a likely culprit? This makes no sense whatsoever. 

Line 177: Again, negative results is a misnomer. Most studies in biology do not have the sample sizes to truly say if something is a negative result. That's why p-value is more highly emphasized then power after running the statistical tests (not to be confused with power analysis during experimental design). 

Line 214: Why would you exclude freshwater species? The field is concerned with the effects of CO2 on behavior. Salinity has no bearing on how blood chemistry is altered, nor how the olfactory epithelium functions. This begs the question of how the "decline effect" results would hold up with the two excellent paper showing olfactory and anxiety in salmon, or the reverse OA effects observed in catfish (both high impact studies). Those were published in 2015 and 2016. At the very least an analysis that includes them should also be included for the readers to evaluate.

Line 231: I wish something was done with the life stage data. One potential biological issue here is a shift away from sensitive life stages to more resilient ones. It is well established that early life stages are more sensitive to a number of environmental issues because the increased surface area to volume ratio. 

Line 320: Should probably be more specific regarding the variance cut off threshold. 

Line 332: "and if temperate species are tolerant". But why would the authors have this expectation? 

Reviewer #2:

I really enjoyed reading this paper. It is clear and very accessible. The research question is also highly current and relevant. The paper make an excellent case study, and I could easily see myself using it in my lectures on research ethics. Below I make some minor comments as well as some suggestion for additional analyses.

Line 15: typo - should be "decreasing"

Line 130 - 131: I agree that bias can skew results, but you don't actually have any evidence to show biased behaviour in your study. Whether patterns are due to confirmation bias would be very difficult to demonstrate. Given this I think you need to think about whether and how you make this point, to make sure it doesn't imply you have evidence showing that experimenter bias was at play. Having said that it might be possible (and would be interesting/relatively easy) to look at whether there is any evidence that effect sizes are greater when studies were conducted blind or not…

Line 142 - 147: I'm not sure I get the purpose of this analysis or that I quite understand what you are trying to get at (see also comment in methods section). I feel like the journal impact factor stuff is also kind of conflated with time since publishing a novel result… I wonder if it would be better to use funnel plots to look for publication bias. To me it feels like what's going on is just general publication bias (i.e. studies with small sample size only get published if there is a significant result and not if it is null) and doesn't necessarily need to invoke journal impact factor (apart from high impact studies being more likely to publish novel results). The fact that the journal impact factor has decreased over time is probably true for any novel result/anything that's published in a high impact journal. I guess I'm just not sure it tells you anything interesting.

Line 150: typo - "fuel"

Line 151: I think this citation bias that might be published by stuff published in high journals is potentially very interesting, and perhaps could be looked into further. It might be quite nice to follow this idea through. Line 153 - I don't see how you sample size analysis tells you anything about how much these papers have been cited… do you mean because high impact journals are cited more? But that doesn't necessarily mean these specific studies are cited more. That is very easy data to get though. Perhaps you could compare citations for papers from different impact factors while controlling for Year or something?

Line 165 & 166: I don't think you show any evidence that supports the claim of observer bias. This early pattern could be solely driven by low sample sizes in combination with publication bias

Line 170: this is a great recommendation, but it is made in hindsight, knowing what is a good sample size... is it possible to make a more general minimum sample size recommendation? Even 30 feels on the low side for a new finding that has never been shown before. Or a call to increase sample sizes more quickly once first study is out, so that the literature is rectified earlier? I was surprised that it took 5 years to start seeing decent sample sizes in the literature... if science is working shouldn't the first thing after seeing a novel result in the literature be to see if the result is robust... Well ideally people might think to do that before publishing a novel result for the first time... but if not then.

Line 177: I'm not really sure how pre-registration ensures all negative results are published in a timely manner. I remember reading (sorry can't remember where) that pre-registered studies often go unpublished. I do see that pre accepting papers based on their methods rather than there results could help with publication bias though.

Line 199: I think that it would be worth being more explicit about what the overall effect size is for your data... Seems to me that for the set of traits you look at there is probably no effect, or at most a very weak effect. You could have a decline effect when the effect of interest is still important and I think it is worth reporting the average effect size for your study, as well as perhaps doing some analyses that demonstrate how much of an influence this effect can have on our overall findings and for how long

Line 208: How many articles were returned by the search. I think it would be good to provide a prisma flow chart for your literature selection http://www.prisma-statement.org/

Line 218: At first when I read this I found it surprising that you managed to go straight from abstract screening to your full list of 95 analysed papers - usually more fall out when you read the methods and results and realise they haven't done what you thought. Reading further on I realise that they did fall out. I think I would report your sample size as the final number of papers that contributed to the meta-analysis (88 I think). Full prisma flowchart would help with this.

Line 232 - 241: I'm not sure of the value of doing this qualitative analysis. What matters is the actual effect sizes no? Is there reason to think these don't match up? Otherwise its just another way of asking the same question… I think you could more usefully use the space to do some of the other suggested analyses

Line 248-249: Do you mean you just took the effect size for the main effect and ignored interactions, or that you left studies that included other factors out all together? I think most likely the former, but the wording is confusing to me

Line 283: Is it possible to include these by transforming the values so they don't cross 0?

Line 366: I'm not sure about this focus on high impact journals here. I wonder if you could just use a funnel plot to show publication bias. I can see why the impact of the journal matters for perception of an effect and citations etc, but for a meta-analysis and estimating an overall effect size it shouldn't matter, and if High profile journals aren't publishing nonsignificant results early on you should still see these results in the literature in low impact journals... it shouldn't really matter. I think trying to get into high impact journals might encourage bad behaviour, but I don't think its high impact journals per se that drives the decline effect.

I don't think showing a relationship between impact factor and time provides evidence of publication bias - that would occur even if there wasn't publication

Reviewer #3:

Title - "ocean acidification fish ecology" is awkward phrasing.

Line 30 - "Ground-breaking" seems too restrictive. Really, it is the first time that a particular hypothesis or the effect of a certain variable is assessed. The effect size in those first studies is large, and declines thereafter. Whether that work was ground-breaking is not the determinant of this, and the decline effect is not restricted to only truly ground-breaking work.

Lines 31 + 32 - it is not really that they fail to replicate the initial effect. It is really that the effect size is smaller. That is a subtle but important distinction.

Line 33 - I am not sure that I can agree that the decline effect itself is what has fueled the reproducibility crisis. It may be one of the drivers, but it is not the only one, and probably not the most important one.

Overall, the opening paragraph needs refinement. Alternately, it could be deleted and you can start at line 34.

Lines 39-40 - what comes after the semi-colon is unnecessary. Recommend deleting it.

Lines 41-42 - should read "using research on THE IMPACT of ocean acidification on fish behaviour" - there are numerous examples through the text where the wording can be improved for clarity.

Line 44 and elsewhere - tone down the word choice from "drastically". Also, the phrase "appear drastically overestimated" is an phraseology contradiction. 

Lines 44 - 45 - do the biases really cause the decline effect? 

Line 45 - "Ways to mitigate the issues…" What ways, and what issues. Either elaborate, or delete.

Line 47 - "Fishy effects" - I realize that you are using a double entendre, but I advise against it. I would argue that this work will reach, and be accepted by, a wider audience if you refrain from this kind of thing and keep the wording tight and not over-stated or too strong and sweeping.

Line 49 - Isn't it a lot more than 300 per year?

Line 53 - I am not sure what a "profound" report is. I come back to word choice. Please be more thoughtful about using appropriate words or phrases to explain what you mean.

Line 54 - Another interesting analysis would be to assess whether such journals publish initial strong effect studies more than other journals.

Lines 55-58 - Consider - would this be more suitable for a mass media article as opposed to a primary research article?

Lines 69-70 - the way that this is stated implies that the strength of the effect was determined by the way that the authors interpreted it. I assume that is not the case and that you are really referring to the statistical effect size? Either way, this should be reworded for better clarity-distinction between the qualitative vs. quantitative analysis.

Line 80 - what is an "outstandingly" large effect size? Please describe how you define it (as opposed to say simply a "large" effect size), or select a more meaningful-accurate word. Perhaps it would just be clearer to say something like the effect sizes of studies after the original ones are almost uniformly smaller?

Line 85 - elaborate on what you mean by "baseline behaviours".

Line 87-88 - they exhibit a decline effect, as opposed to being characterized by one. Reconsider the need to say "strongly" - the reader can see it in the Figure.

Lines 110-111 - selective exclusion of data is very hard to document and assess. You might consider making it clear that you do not have such evidence, you are just making a general comment.

Lines 130-135 - If you do not have evidence to document any of this, then make clear that you are speculating, albeit reasonably, about some of the likely drivers of the decline effect.

Line 143-147 - not surprising, but good to see it quantified.

Line 150 - spelling problem here - not sure what "can fue citation bias" means?

Line 165 - although it is a reasonably high probability, you do not actually provide any evidence of experimenter bias, so I would not overstate that.

Line 166 - what is an "outstanding" effect?

Line 175 - I am not sure that it would ever be possible to achieve completely unbiased results. Perhaps rephrase, "to limit biases"?

Line 177 - pre-registration does not on its own eliminate publication bias.

Lines 186-189 - I find that this is another example of making an unfounded accusation (albeit a likely true one) that will not serve to help your case and which is unnecessary. If you have evidence of it, you can say it. Otherwise, you can just say that editors should use a diversity of reviewers.

Line 198-199 - I recommend that you delete this and start with "Our data demonstrate…"

Methods. I am not an expert on meta-analysis methodology so I defer to other reviewers who are.

Lines 219-200 - the latter part of this sentence is unnecessary.

Lines 231-241 -It seems to me that it would have also been interesting to compare the quantitative effect sizes with those that you extracted from the discussion (for the same article)?

Reviewer #4:

This meta-analysis is an exciting and important work showing a decline effect. However, several aspects of methodological procedures and analyses can be much better. I believe that this MS needs to be improved to warrant a place in PLoS Biology, because this meta-analysis (systematic review) will be influential work so that it should have a high standard. Also, it does not cite some previous key papers, which I mention below. 

1. It requires better reporting of search results, inclusion/exclusion criteria etc, which are detailed in PRISMA reporting guideline that is used for many systematic reviews. I would at least like to see a PRISMA diagram showing the results of screening processes. One can see an example of this in a very relevant work:

Sánchez-Tójar A, Nakagawa S, Sanchez-Fortun M, Martin DA, Ramani S, Girndt A, Bókony V, Kempenaers B, Liker A, Westneat DF, Burke T. Meta-analysis challenges a textbook example of status signalling and demonstrates publication bias. Elife. 2018 Nov 13;7:e37385.

2. Sanchez-Tojar et al.'s paper shows the importance of grey literature, which is also mentioned in the PRISMA explanation & elaboration. There is no mention of the grey literature search - the authors did use Google Scholar, which does include grey literature. 

3. the authors used absolute values for meta-analysis, and as far as I can see, wrong sampling variances (meta-analytic weights) were used for meta-analysis. Please see this paper:

Morrissey MB. Meta‐analysis of magnitudes, differences and variation in evolutionary parameters. Journal of Evolutionary Biology. 2016 Oct;29(10):1882-904.

AS this paper will show, the current analysis is incorrect. So I recommend using the transform-and-analyze approach. 

4. the authors dichotomized continuous variables (e.g., strong effect, N > 30 etc). I strongly advise against such an approach (leading to Type I errors). Please see this paper:

Royston, Patrick, Douglas G. Altman, and Willi Sauerbrei. "Dichotomizing continuous predictors in multiple regression: a bad idea." Statistics in medicine 25.1 (2006): 127-141.

5. I would recommend using the better version of lnRR (small sample size corrected one) proposed by this paper:

Lajeunesse MJ. Bias and correction for the log response ratio in ecological meta‐analysis. Ecology. 2015 Aug;96(8):2056-63.

6. There are several key relevant papers missing. For example:

Jennions MD, Møller AP. Relationships fade with time: a meta-analysis of temporal trends in publication in ecology and evolution. Proceedings of the Royal Society of London. Series B: Biological Sciences. 2002 Jan 7;269(1486):43-8.

Koricheva, Julia, and Elena Kulinskaya. "Temporal instability of evidence base: a threat to policy making?." Trends in ecology & evolution 34.10 (2019): 895-902.

Murtaugh PA. Journal quality, effect size, and publication bias in meta‐analysis. Ecology. 2002 Apr;83(4):1162-6.

---

## [Decision Letter · Decision Letter 2]

4 Nov 2021

Dear Jeff,

Thank you for submitting a revised version of your manuscript "An extreme “decline effect” in ocean acidification effects on fish behaviour" for consideration as a Meta-Research Article at PLOS Biology. This revised version of your manuscript has been evaluated by the PLOS Biology editors, the Academic Editor and the original reviewers.

In light of the reviews (below), we are pleased to offer you the opportunity to address the remaining points from the reviewers and the Academic Editor in a revised version that we anticipate should not take you very long. We will then assess your revised manuscript and your response to the reviewers' comments and we may consult the reviewers again.

IMPORTANT: Please address the following:

a) Please attend to the remaining requests from reviewer #1.

b) We are aware that some of you harbour suspicions about the scientific validity of two key studies that are analysed here (e.g. https://www.science.org/content/article/does-ocean-acidification-alter-fish-behavior-fraud-allegations-create-sea-doubt). Indeed, that Science article cites the preprint of the current submission. Given this, the Academic Editor has made the following request: "The authors should do a formal re-analysis without those two studies. I don't really get how this is a decline effect if it is because of misconduct rather than poorly designed experiments or simply a regression to the mean effect. In other words, is "misconduct" a plausible mechanism for the decline effect, or does it have to be something else? In which case, the leverage of those two studies on the meta-analysis seems particularly important."

c) Please address my Data Policy requests below; specifically, please supply numerical values underlying Figs 1ABCD, 2ABC, 3, 4ABCDE, and cite the location of the data clearly in each relevant Fig legend (I note that the raw data and code are already provided, but we’ll need the above output values too). These can be included as supplementary data files (S1_Data.xlsx, etc.) or in a public repository such as Dryad, Github, Figshare, etc. 

We expect to receive your revised manuscript within 1 month.

**IMPORTANT - SUBMITTING YOUR REVISION**

*Resubmission Checklist*

*Published Peer Review*

*PLOS Data Policy*

*Blot and Gel Data Policy*

Sincerely,

Roli

Roland Roberts

Senior Editor

PLOS Biology

rroberts@plos.org

REVIEWERS' COMMENTS:

Reviewer #1:

I would like to thank the authors for seriously considering all of the reviewer comments. I found this manuscript to be greatly improved from the prior version. I must also say that the authors convinced me that one of my concerns was clearly misplaced. Specifically, the premise that a few early papers had an outsized impact on the observed effect, and that these papers were no longer held up as representative of the field. I was very surprised to see that these papers are still receiving 60-80 citations a year! I have a few minor suggestions for the authors to consider, but otherwise I think the authors have done a nice job revising this work. 

#1) The references to citations (beginning on line 201 and Figure 4) are very useful, but Google Scholar was a curious choice. Considering this is being used to compare to impact factor, I think that ISI citations are a much better and more accurate measure of the peer reviewed literature. 

#2) Line 259-268: Recent events, which I will not detail here, have put the ocean acidification decline effect in a different light. I agree with the authors points about scrutinizing methods, best practices, data transparency and general skepticism, but I wonder if the publishing suggestions should be reconsidered. I don't think registered reports or pre-acceptance of approved methods would have had any positive impact on this particular situation. To me, the authors hit it on the head with the skepticism comment. I would only add that the scientific community was also slow to absorb the published negative results, which are clearly cited less than the initial studies. I would hope this is not the fault of people directly in the area, who are up to date on the newest work. My guess is many of these citations come from ancillary fields that cite high impact work in broad terms. I don't have a solution to that problem, but it seems that it is definitely more the fault of scientists not being thorough in their literature research than it is the fault of the publishing system. 

Reviewer #4:

The authors have very carefully addressed my comments and also the comments of other reviewers. It will be a very important contribution!

---

## [Editor Report · Decision Letter 3]

26 Nov 2021

Dear Jeff,

Thank you for submitting your revised Meta-Research Article entitled "An extreme “decline effect” in ocean acidification effects on fish behaviour" for publication in PLOS Biology. I have now assessed your responses and revisions and discussed them with the Academic Editor. 

Based on our assessment, we will probably accept this manuscript for publication, provided you satisfactorily address the following editorial, data and other policy-related requests.

IMPORTANT: Please attend to the following:

a) We're struggling with the Title a bit. One problem is that "decline effect" is not a widely understood phrase; the second is the duplication of the word "effect," and the third is the absence of the method used; overall the title is currently quite opaque. We suggest something like "Systematic review and meta-analysis suggests that ocean acidification has a negligible direct impact on fish behaviour," if you feel happy with that; however, I'm open to discussing alternatives.

b) After some debate, and partly because of the specific relevance to this field, we think that it would be more appropriate to publish your paper as a regular Research Article, rather than a Meta-Research Article. This has no formatting implications, but please can you change the Article Type to "Research Article" when you resubmit?

c) As this is a systematic review and meta-analysis, do check that you've complied with our policy by providing a completed PRISMA checklist and flow diagram. I see that Fig 6 is a flow diagram, but please check our policy here: https://journals.plos.org/plosbiology/s/best-practices-in-research-reporting#loc-reporting-guidelines-for-specific-study-types

d) Please address my Data Policy requests below; specifically, please supply numerical values underlying Figs 1ABCD, 2ABC, 3, 4ABCDE, and cite the location of the data clearly in each relevant Fig legend (I note that the raw data and code are already provided, but we’ll need the above output values too).

e) Because of a slight nervousness about the possible reaction of Munday, Dixson and co, I've been asked to refer this paper up the management chain, just to check that we don't need legal counsel. I personally think that you've worded everything in a scholarly and appropriately cautious manner; for anything stronger, the reader is merely referred to Enserink's Science article. However, I just thought I'd give you a heads-up just in case the powers-that-be come back and want something tempering.

We expect to receive your revised manuscript within two weeks. 

*Published Peer Review History*

*Early Version*

Sincerely,

Roli

Senior Editor,

rroberts@plos.org,

PLOS Biology

DATA POLICY:

Many thanks for providing the raw data and code. However, we also need the numerical values that underlie the figures and results of your paper be made available in one of the following forms:

Regardless of the method selected, please ensure that you provide the individual numerical values that underlie the summary data displayed in the following figure panels as they are essential for readers to assess your analysis and to reproduce it: Figs 1ABCD, 2ABC, 3, 4ABCDE. NOTE: the numerical data provided should include all replicates AND the way in which the plotted mean and errors were derived (it should not present only the mean/average values).

DATA NOT SHOWN?

---

## [Editor Report · Decision Letter 4]

8 Dec 2021

Dear Jeff,

On behalf of my colleagues and the Academic Editor, Andrew Tanentzap, I'm pleased to say that we can in principle accept your Research Article "Meta-analysis reveals an extreme “decline effect” in the impacts of ocean acidification on fish behaviour" for publication in PLOS Biology, provided you address any remaining formatting and reporting issues. These will be detailed in an email that will follow this letter and that you will usually receive within 2-3 business days, during which time no action is required from you. Please note that we will not be able to formally accept your manuscript and schedule it for publication until you have any requested changes.

PRESS: We frequently collaborate with press offices. If your institution or institutions have a press office, please notify them about your upcoming paper at this point, to enable them to help maximise its impact. If the press office is planning to promote your findings, we would be grateful if they could coordinate with biologypress@plos.org. If you have not yet opted out of the early version process, we ask that you notify us immediately of any press plans so that we may do so on your behalf.

Best wishes,

Roli 

Roland G Roberts, PhD 

Senior Editor 

PLOS Biology

rroberts@plos.org